# Environmental Effects of Credit Allocation Structure and Environmental Expenditures: Evidence from China

**Qiming Yang [1], Jun He [1,\*], Ting Liu [1] and Zhitao Zhu [2]**

[1] School of Management, University of Science and Technology of China, Hefei 230026, China; yqmsir@mail.ustc.edu.cn (Q.Y.); tingliu@ustc.edu.cn (T.L.)

[2] Institute of Advanced Technology, University of Science and Technology of China, Hefei 230026, China; andyzzt@mail.ustc.edu.cn

\* Correspondence: hejun@ustc.edu.cn

**Abstract:** This article studies how the allocation structure of bank credit capital between state-owned and private enterprises and government environmental expenditures affect environmental pollution in China. The present literature argues that credit allocation and government environmental expenditures may play an important role in environmental quality improvement. However, these studies rarely consider the credit allocation structure between State-owned enterprises (SOEs) and private enterprises; in addition, they overlook the interaction effects of credit allocation and government environmental expenditures. Based on these, we put forward three hypotheses. Moreover, the study applies relevant spatial data for 2011–2017 from 31 provinces in China to a spatial econometric model, and the results indicate that (1) environmental pollution among provincial regions shows a significant positive spatial autocorrelation. (2) Environmental expenditures and environmental pollution display an inverse U-shaped relationship, which supports the numerical simulation results. (3) The interaction effect of credit allocation structure and environmental expenditures on environmental pollution is significantly positive, which means that the allocation of more credit capital to private enterprises will restrain the effect of government environmental expenditures. With the increasing significance of environmental protection in China, it is necessary to strengthen the supervision of private enterprises' environmental pollution behavior, expand government expenditures on ecological protection, and promote regional collaborative environmental governance to improve environmental quality.

**Keywords:** environmental pollution; empirical research; spatial econometric; China

## 1. Introduction

After decades of rapid economic growth, development in China has achieved outstanding results. However, environmental problems caused by economic development have gradually emerged [1]. According to the Environmental Performance Index: 2018 Report, China ranks 120th, and the country's air quality ranks fourth from last internationally. Data from the National Bureau of Statistics of China show that the average annual growth rate of China's government environmental expenditures exceeded 13%, which is much higher than China's average annual GDP growth rate from 2011 to 2019. Environmental pollution and the high cost of pollution control have already affected the economy's high-quality development [2].

In the modern economic system, bank credit resources are an important support for industrial development, and enterprises are highly dependent on external financing sources [3]. However, the scarcity of credit resources has led to competition for credit resources among enterprises. SOEs constitute a key part of economically dominant industries [4]. Under the political connections between SOEs and the government, SOEs can obtain implicit guarantees from the government, which reduces the performance cost of debt contracts, thereby gaining more credit resources. Due to a lack of credit resource support, private enterprises may reduce environmental investment to maintain operations and

expand production, while SOEs are subject to stricter environmental protection regulations due to government influences, and the willingness to protect the environment is stronger. In recent years, some studies have focused on the relationship between China's credit allocation structure and environmental pollution; however, the results are limited, and they mainly focus on the credit allocation structures among various industries rather than those between state-owned and private enterprises. With the increasing environmental protection awareness, China's government is attaching an important role to environmental protection [5], and various instruments of environmental regulation and investment have been used to improve environmental quality [6]. Government environmental expenditure will be an important measure for the government carry out the environmental performance in the long run [7].

The present work differs from the current literature in the following ways. First, by processing the loan data of Shanghai and Shenzhen A-share listed companies, we reveal the environmental impact of the allocation of credit resources between SOEs and private enterprises, which differs from other studies that focus on the credit allocation of various industries. Second, we introduce the interaction terms of the SOEs and government environmental expenditures, and examine the impact of the interaction between the two on environmental pollution. Third, we construct a comprehensive environmental pollution index using four types of pollutants as an indicator variable of environmental pollution, thus diverging from most studies that use a single pollutant, such as $SO_2$, as an indicator variable of environmental pollution.

The remainder of this study proceeds as follows. Section 2 highlights the literature review and hypothesis development. Sections 3 and 4 present the empirical strategy and data, and results, respectively. Section 5 discusses the result and gives the policy suggestions.

## 2. Literature Review and Hypothesis Development

### 2.1. Credit Allocation Structure and Environmental Pollution

Existing research rarely discusses the relationship between credit allocation structure and environmental pollution directly, though some research on credit misallocation and sustainable finance provides guidance in this area. Generally, credit resources are an important support for industrial development and production activities, and the impact of credit allocation structure on the environment is often related to the industrial structure in place. China's extensive industrial development model, attained at the cost of natural resources and environmental quality, has led to an imbalance in credit allocation structure: credit resources were concentrated among large enterprises and other popular industries in past decades through what some researchers called capital misallocation [8,9], and the steel, coal, chemical, and other heavy industries attracted the majority of bank credit resources [10]. Energy-intensive industries in China consume a large amount of energy in production process [11,12]. These energy-intensive industries also have related problems such as heavy pollutant emissions, blind expansion of production scale, and low profitability [13,14]. Moreover, some research focuses on sustainable finance. A sound bank credit resource allocation system can promote enterprise innovation and industrial upgrading by improving the efficiency of credit allocation [15], which means that the credit allocation structure and the social responsibility of environmental protection are interrelated. Banks can encourage enterprises to assume social responsibility for environmental protection through the adoption of differentiated credit allocation policies [16]. Now we are focused on the credit allocation between SOEs and private enterprise, and the different effects they have on the environment. Inspired by the growth miracle of East Asian countries, a large body of literature has supported that incentive industrial policies are important for enterprises to invest in long-term assets and innovation activities [17–19]. China has introduced credit policies to encourage the development of small and medium enterprises, most of which are private enterprises. Generally, private enterprises have faced the demand of their owners for the possibility of profit [20]; they have little inclination toward environmental investment or pollutant treatment because of the extra cost. On the other hand, alongside

the capacity for production, transactions, and other economic activities, SOEs are politically and administratively controlled by government [21,22]. In order to reduce the pollutant emissions and satisfy people's demand for environmental improvement, the government has adopted stricter environmental regulations for SOEs. In the mid-2000s, China's government has incorporated some SOEs into its strategic agenda on reducing pollutant emission and improving the energy efficiency, and empirical studies have suggested that SOEs accomplish the government's targets satisfactorily [23–25]. Compared with private enterprise, SOE managers have higher levels of self-reported environmental ethical values [26]. Based on the different business philosophies and social responsibilities between SOEs and private enterprises when they face the conflict of economic expectations and environmental goals, we thus propose the following hypothesis.

**Hypothesis 1 (H1).** *The more credit resources private enterprise is allocated, the more serious the environmental pollution.*

*2.2. Government Environmental Expenditures and Environmental Pollution*

As the main governor and regulator of environmental pollution, the government plays an important role in the treatment of environmental pollution. Related research mainly focuses on the relationship between government public expenditures and environmental quality, but no unanimous result has been reached. Some studies indicated that government public expenditures have external effects on production activities and that there is a positive correlation with pollutant emissions [27,28]. Basoglu et al. analyzed the effect of environmental expenditures made by the public sector on the ecological deficit in nine coordinated market economies in Europe from 1995 to 2014; the study indicated that the scale effect of public expenditures affects environmental quality negatively [7]. Some other studies have found a negative correlation between government public expenditures and environmental pollution [29–31]. Government environmental expenditures are a part of government public expenditures that can guide the direction of social investment and constrain the environmental protection behavior of enterprises, which has a positive impact on environmental protection [32]. Gholipour et al. analyzed the carbon dioxide and PM10 data of 14 MENA(Middle Eastern and Northern Africa) countries and found a cointegration relationship between government environmental expenditures and air quality and that environmental expenditures can improve air quality [33]. He et al. found a long-term relationship between environmental expenditures and air quality through an empirical study of air quality panel data for some Chinese cities, and their results showed that an increase in environmental expenditures of 1% can improve air quality in some cities by 0.051% [34]. Although most studies have argued that government environmental expenditures, as part of government public expenditures, have a positive impact on environmental quality, these studies rarely considered the fact that the externalities of government public expenditures on production activities may increase environmental pollution.

Actually, a literature review showed that the effects of different government expenditure structures on environmental quality vary. Basoglu et al. indicated that highlighting the environmental expenditures can boost the welfare of the environmental quality instead of the scale of the public expenditures [7]. Lu et al. examined the influence of China's fiscal expenditure structure on environmental pollution and noted that pollution would be exacerbated when noneconomic expenditures increase and it would be reduced when the environmental regulation effect dominates [35]. He et al. believed that the increase in the proportion of economic construction expenditures would aggravate China's environmental pollution, while increasing the proportion of social service expenditures would be conducive to reducing environmental pollution [34]. Environmental quality expands according to the government's environmental expenditures, which is consistent with literature. However, government's non-environmental expenditures restrict the role of environmental expenditures in improving environmental quality; moreover, the impact they make on environmental quality may be negative. Hence, we propose the following hypothesis:

**Hypothesis 2 (H2).** *The effect of government environmental expenditures may be constrained by other government public expenditures, and there is a non-linear relationship between the relative scale of environmental expenditures and environmental quality.*

*2.3. The Interaction Effect of Credit Allocation Structure and Environmental Expenditures*

Government environmental expenditures are mainly used to control environmental pollution, while enterprises expand production through external financing, such as bank credit, and generate more pollutants. Government environmental expenditures and bank credit resources seem to play completely different roles in environmental pollution, but in fact, there are close ties between them. In most countries, government control of financial institutions, especially banks, is a very common phenomenon [36], and 40% of the world's banks are controlled by governments according to statistics from the World Bank. In a perfectly competitive market, various resources are usually allocated efficiently; however, when the market regulation function fails, a government can adjust the allocation structure of credit resources through banks under its control [37]. In addition, credit resources are highly complementary to government expenditures and social investments. The green credit policy implemented by banks optimizes the allocation of corporate credit resources, promotes the adjustment of corporate industrial structure, and improves the ecological environment [38–40]. The above studies show a connection between bank credit resources and government expenditures and that both affect environmental quality to varying degrees. However, existing studies have not considered the joint impact of government expenditures and credit allocation structure on environmental pollution. So the following hypothesis is proposed:

**Hypothesis 3 (H3).** *Credit Allocation Structure and Environmental Expenditures may have an interaction effect on environmental pollution.*

## 3. Materials and Methods

### 3.1. Sample Selection

The study selected the panel data set of 31 provinces in China from 2011 to 2017 as a sample. We chose 2011 as the starting year because China's government has set "energy conservation and environmental expenditures" since 2011. In addition, the data set on pollutant emission was selected from the China Pollution Source Survey, and the latest survey updated the data up to 2017. So our data set ends in 2017. Most of the dataset was drawn from the National Data, National Bureau of Statistics of China.

The data set on credit allocation structure was assembled from two samples for empirical research and robustness test. The first was from above-scale industrial enterprises, which are divided into SOEs and private enterprise, and the data set was selected from the National Data, National Bureau of Statistics of China. The second was drawn from A-share companies listed on the Shanghai and Shenzhen stock exchanges, excluding real estate companies, financial companies, and special treatment (ST) companies. Then we divided them into SOEs and private enterprise by their ownership. The data set was selected from the Wind Economic Database.

### 3.2. Variables

Pollution Indicator: Comprehensive Index of Environmental Pollution (P). As multiple dimensions of environmental pollution are caused by waste discharge from production and household activities, the selection of a single pollutant as an environmental pollution indicator is insufficient to demonstrate the comprehensive attributes of environmental pollution. Therefore, four categories of pollutants with strong spatial mobility and representativeness were adopted as environmental pollution indicators in this paper. Appendix A shows the calculation method of the Comprehensive Index of Environmental Pollution (P). The specifically selected discharge indicators considered include industrial sulfur dioxide emissions, industrial nitrogen oxide emissions, industrial soot emissions, and industrial effluent discharge.

Independent Variables: (1) Credit Allocation Structure (CAS). Credit allocation structure in this paper represents the allocation of bank credit resources between private enterprises and SOEs. The net value of fixed assets in all industries is ultimately determined by the industrial value added. Thus, it was possible to utilize the ratio of loan balances of nonstate-owned industrial enterprises to industrial enterprises' loans approximately as a substitution variable. As China's bank interest rates are governed by the Central Bank's interest rate policy, there are minor fluctuations of bank interest rates in the short term. In the meantime, in consideration of data availability, this paper adopted the ratio of interest expenditures of above-scale nonstate-owned industrial enterprises to that of above-scale industrial enterprises to evaluate credit allocation structure. (2) Governmental Environmental Expenditures Level (EXP). Within China's fiscal budget system, the environmental expenditures category was introduced in 2007, and energy conservation expenditures items were incorporated in 2008. As a result, the energy conservation and environmental expenditures category was officially established in 2011. In this paper, the ratio of energy conservation and environmental expenditures to total annual budget expenditures was employed to determine the value of government environmental expenditures. (3) Interaction Term (CAS_C*EXP_C). This value was calculated as the product of centralized credit allocation structure and government expenditures data. (4) Credit Allocation Structure Substitution Variable (CAS2). CAS2 was treated as a substitution variable for CAS in the robustness test. In this paper, some of the A-share companies listed on the Shanghai and Shenzhen stock exchanges were selected as samples by province. The remaining companies were classified into SOEs and private enterprises by ownership. The ratio of total long- and short-term loans of private enterprises to those of SOEs was used to assess the credit allocation structure, which was denoted as CAS2.

Control Variables: Control variables were introduced into the regression model to minimize the estimation bias induced by the omission of variables. Consolidating the existing literature [41,42], the following other variables that influence environmental pollution were selected as control variables: (1) Investment Rate (INV). Pollutants are the by-products of production and investment. An increased investment rate will intensify pollutant emissions and energy consumption, thus creating environmental pollution. This paper measured the investment rate from the proportion of fixed asset investment to provincial GDP. (2) Technology Innovation Expenditures (INN). Technology advancement can be promoted by the government or by enterprises through increased expenditures on technological innovation, which can in turn contribute to cleaner and more environmentally friendly production activities, reducing environmental pollution. The ratio of the sum of science and technology expenditures and education expenditures in the government expenditures budget to the total expenditures budget was adopted as an indicator of technology innovation expenditures. (3) Industrial Structure (INDU). Following the process of the industrial revolution, there has also been an increasing level of global environmental pollution. In the early stages of industrialization, the development of extensive industry generated excessive resource exploitation and pollutant emissions. An increase in the proportion of industrial output was accompanied by an intensified level of environmental pollution. Since a certain degree of economic development has been reached, with the further optimization of the industrial structure, the proportion of the tertiary industry, which is more environmentally friendly, has increased, alleviating environmental pollution. Hence, this paper adopted the proportion of the output value of secondary industry to the total provincial output value to weigh the industrial structure. (4) Urbanization Level (CITY). As major hubs for industry and population, urban regions demand more resources for production and consumption and generate more pollutants than rural regions. Increasing urbanization in China each year has resulted in more prominent environmental concerns emerging in urban regions. Urbanization level was measured by the proportion of the urban population to the total provincial population in this paper. (5) Economic Development Level (PGDP). Environmental pollution is a necessary cost of economic development. The development of various industries and domestic production accounting are ultimately embodied in GDP where

the larger the GDP value is, the higher the relative environmental pollution level will be. In this paper, GDP per capita was adopted to measure the level of economic development.

Considering data availability, data of 31 provincial administrative entities in China from 2011 to 2017 were selected as our sample. The data were sourced from the *China Statistical Yearbook*, EPSDATA database and the Wind database. The descriptive statistics of the variables are summarized in Table 1.

**Table 1.** Descriptive statistics of variables.

| Variable | Full name | Mean | Maximum | Minimum | Standard Deviation | Observation |
|---|---|---|---|---|---|---|
| P | Pollution | 0.0323 | 0.0456 | 0.0239 | 0.005 | 217 |
| CAS | Credit Allocation Structure | 0.4704 | 0.8789 | 0.0239 | 0.1822 | 217 |
| EXP | Governmental Environmental Expenditures | 0.0289 | 0.0672 | 0.0118 | 0.0091 | 217 |
| CAS2 | Credit Allocation Structure 2 | 0.6159 | 2.5789 | 0.0234 | 0.5844 | 217 |
| INV | Investment Rate | 0.8064 | 1.5070 | 0.2366 | 0.2528 | 217 |
| INN | Technology Innovation Expenditures | 0.1842 | 0.2508 | 0.1058 | 0.032 | 217 |
| INDU | Industrial Structure | 0.4493 | 0.5905 | 0.1901 | 0.0838 | 217 |
| CITY | Urbanization Level | 0.5558 | 0.8961 | 0.2273 | 0.1338 | 217 |
| PGDP | Economic Development Level | 5.0049 | 12.9042 | 1.6437 | 2.3388 | 217 |

### 3.3. Model Specification

In general, credit allocation structure and government environmental expenditures have a direct or indirect impact on regional environmental quality via production activities. Based on the test process of the environmental Kuznets curve, credit allocation structure (CAS) and environmental expenditures (EXP) are introduced into the following econometric model Equation (1):

$$P_{it} = \alpha_0 + \alpha_1 FIN_{it} + \alpha_2 EXP_{it} + \alpha_3 EXP^2_{it} + \alpha_4 X_{it} + \varepsilon_{it} \tag{1}$$

where $i$ represents the province, $t$ denotes time, $P$ is the environmental variable, $X$ represents other control variables, and $\varepsilon$ is a random error term.

Beyond production activities, the level of environmental pollution in a region is also subject to that of the surrounding area. Judging from the natural attributes of major pollutants, pollutants such as effluents and gas emissions are highly spatially mobile. In incorporating social attributes such as industrial structure layout and regional energy consumption structure, environmental pollution is generally perceived to exhibit a strong spatial autocorrelation [43]. Hence, we used an empirical spatial econometric approach to explore the relationships between environmental pollution, credit allocation structure, and environmental expenditures.

As there may be different manifestations of spatial dependence of the explained variables, the spatial autoregressive model (SAR) and spatial error model (SEM) were applied. The representations of the two models are presented below:

SAR Model: $Y = \rho WY + X\beta + \varepsilon$

SEM Model: $Y = X\beta + u, u = \lambda Wu + \varepsilon$

where W is an $n \times n$ spatial weight matrix with $n$ individuals, and W depicts the spatial relationship of sample units. We used a spatial adjacency matrix where $W_{ij}$ is taken as 1 when provinces $i$ and $j$ are adjacent and 0 when provinces $i$ and $j$ are not adjacent. WY denotes the spatial interaction effect between the explained variables; Wu represents the spatial correlation between unpredictable random shocks; $\alpha$, $\beta$ and $\lambda$ are coefficients to be estimated; and $\varepsilon$ is the random error term.

### 3.4. Spatial Correlation Analysis and Model Selection

Prior to the regression analysis, the spatial correlation test of the panel data was first performed. For cross-sectional data, it is common to use methods such as Moran's I test for

spatial autocorrelation testing, yet panel data were used in this paper. The traditional spatial correlation test was no longer applicable. Kronecker product $C = I_T \otimes W$, which utilizes spatial weight matrix W and T-dimensional time unit matrix $I_T$, substituted for the spatial weight matrix in the original cross-sectional spatial correlation test. The cross-sectional spatial correlation test, the Moran test, was extended to the panel data with T being the data's time span and T = 7 in this paper. To determine the robustness of the test, apart from Moran's I test, four other tests, namely, Walds, Lratios, LMsar, and LMerr, were also adopted. The test results are presented in Table 2.

**Table 2.** Spatial correlation test.

| Test | Without Interaction Terms | | With Interaction Terms | |
|------|:-----------:|:-----------:|:-----------:|:-----------:|
| | **Statistics** | ***p*-Value** | **Statistics** | ***p*-Value** |
| Moran | 0.203 | 0 | 0.166 | 0 |
| Walds | 341.001 | 0 | 300.197 | 0 |
| Lratios | 41.008 | 0 | 33.101 | 0 |
| LMsar | 102.012 | 0 | 62.755 | 0 |
| LMerr | 17.684 | 0 | 11.797 | 0.0006 |

In light of the results shown in Table 2, the statistics of the five tests were positive and rejected the original hypothesis to at least the 1% significance level, irrespective of the inclusion of interaction term CAS_C*EXP_C, suggesting the presence of a positive spatial correlation for interprovincial environmental pollution in China and the need to use spatial econometric models.

The spatial autocorrelation test already evidenced the existence of spatial dependence between environmental pollution and its influencing factors. In contrast to the SAR and SEM models that introduced spatially lagged explained variables or spatially lagged random error terms in Equation (1), the models no longer complied with the classical hypothesis of the OLS regression model. If an OLS estimation of the spatial panel model was still performed, it would yield either biased or invalid estimation results. Hence, this paper applied the maximum likelihood method (ML) for parameter estimation to the SAR and SEM models. We investigated the spatial effects between environmental pollution and its influencing factors across provinces and territories in China. In general, when regression analysis is confined to some specific individuals, it is more appropriate to adopt fixed effects [44,45]. Consequently, a spatial lag fixed-effects model and spatial error fixed-effects model were employed for regression analysis. In accordance with the different controls over spatial and temporal effects, spatial econometric models can be categorized into four types: nonfixed effects (nonF), spatial fixed effects (sF), temporal fixed effects (tF), and spatiotemporal dual fixed effects (stF).

The selection of SAR and SEM models remained a critical issue. Anselin et al. proposed a model discriminant criterion that is currently well accepted: Without taking into account the constraint of spatial correlation, the model is estimated by the OLS method followed by a spatial correlation test [46]. A stronger explanation for the sample by the SEM model (SAR model) is evidenced if Robust LMerr (Robust LMsar) is more significant than Robust LMsar (Robust LMerr). By applying the above method and comparing the test results, it was found that Robust LMerr (20.7078) and Robust LMsar (7.611) passed the 1% significance test in the model excluding the interaction term. In contrast, Robust LMerr was more significant; in the model including the interaction term, Robust LMerr (7.1727) passed the 1% significance test, whereas Robust LMsar (0.7504) failed the 10% significance test. Therefore, the SEM model was more appropriate for the sample used in this paper, implying that interprovincial environmental pollution variation in China largely arises from unobservable random disparities among individuals.

## 4. Result

### 4.1. Empirical Analysis of Spatial Econometrics without Interaction Terms

The statistical results shown in Table 3 reveal significant positive estimations of ρ and λ, showing positive spatial dependence between interprovincial environmental pollution effects as well as the spatial aggregation of environmental pollution effects between neighboring provinces. A further analysis identified a significant difference between the estimates of the two goodness-of-fit $R^2$ and $corr^2$ values in the spatial fixed-effects and spatiotemporal fixed-effects models with the results for $corr^2$ converging to zero. This finding suggested that the control spatial fixed effects and temporal fixed effects did not fit the sample better, and therefore, these two regression results were eliminated.

**Table 3.** Spatial econometric results (excluding interaction terms).

| Variable | SAR Model | | | | SEM Model | | | |
|---|---|---|---|---|---|---|---|---|
| | nonF | sF | tF | stF | nonF | sF | tF | stF |
| constant | 0.0058 (1.18) | | | | 0.0185 *** (4.47) | | | |
| CAS | 0.0046 ** (2.42) | −0.0006 (−0.52) | 0.0012 (0.66) | −0.0006 (−0.43) | 0.0063 *** (3.35) | −0.0005 (−0.43) | 0.0045 ** (2.42) | −0.0005 (−0.38) |
| EXP | 0.5164 *** (3.47) | −0.0736 ** (−2.04) | 0.3908 *** (2.81) | −0.0768 ** (−2.09) | 0.3891 *** (3.15) | −0.0584 * (−1.71) | 0.3693 *** (2.91) | −0.0587 * (−1.69) |
| EXP² | −6.4765 *** (−3.01) | 0.9078 * (1.91) | −4.7847 ** (−2.37) | 0.9338 * (1.95) | −5.2864 *** (−2.94) | 0.7462 (1.64) | −4.8997 *** (2.66) | 0.7443 (1.61) |
| INV | −0.0034 ** (−2.26) | 0.0002 (0.38) | −0.011 *** (−6.04) | 0.0002 (0.333) | −0.0107 *** (−6.69) | −0.0001 (−0.0004) | −0.0123 *** (−7.23) | −0.0001 (−0.03) |
| INN | 0.0482 *** (4.61) | 0.0054 (1.21) | 0.049 *** (4.95) | 0.0081 (1.45) | 0.0567 *** (5.64) | 0.0072 (1.44) | 0.0536 *** (5.38) | 0.0081 (1.44) |
| INDU | 0.0184 *** (5.28) | 0.0018 (1.01) | 0.0301 *** (8.11) | 0.0011 (0.38) | 0.0215 *** (6.61) | 0.0016 (0.81) | 0.0261 *** (7.74) | 0.0005 (0.17) |
| CITY | −0.0115 ** (−2.35) | 0.0019 (0.53) | −0.0141 *** (−3.09) | 0.0025 (0.41) | −0.0109 ** (−2.49) | 0.0034 (0.85) | −0.0121 *** (−2.74) | 0.0058 (0.88) |
| PGDP | 0.0003 (1.01) | 0.0001 (0.74) | −0.0002 (−0.85) | 0.0001 (0.86) | −0.0004 (−1.49) | 0.0001 (0.66) | −0.0005 ** (−1.97) | 0.0002 (0.92) |
| ρ | 0.1799 ** (2.37) | 0.3299 *** (4.11) | 0.2429 *** (3.51) | 0.3983 *** (5.24) | | | | |
| λ | | | | | 0.6359 *** (11.131) | 0.3539 *** (4.44) | 0.4709 *** (6.58) | 0.4117 *** (5.42) |
| R² | 0.4088 | 0.9854 | 0.5109 | 0.9857 | 0.2181 | 0.9837 | 0.4541 | 0.9839 |
| corr² | 0.3643 | 0.0634 | 0.4604 | 0.0798 | 0.3229 | 0.0462 | 0.461 | 0.0578 |
| Log L | 897.359 | 1295.686 | 916.458 | 1297.613 | 915.58 | 1295.309 | 924.883 | 1296.9386 |

Note: *, **, and *** indicate that significance tests at the 10, 5, and 1% levels were passed, respectively, and the t-statistic values are shown in parentheses.

In the SEM model controlling for temporal fixed effects, there was a significant positive coefficient on the credit allocation level (CAS), implying a significant impact of the credit allocation structure between private enterprises and SOEs on interprovincial environmental pollution in China. In particular, environmental pollution levels were lower when SOEs occupied more bank credit resources and vice versa. Two main reasons for this finding are as follows. On the one hand, SOEs are under the direct control and management of the government, whose enforcement of environmental regulations is stronger, and they are thus subject to more stringent supervision. Compared to private enterprises, the pollutants produced by SOEs through their production activities can be effectively controlled and treated. More credit support is available to SOEs such that environmental pressure is relatively low; on the other hand, owing to the profit-seeking nature of capital, in times of incomplete and non-standardized regulatory intensity and environmental norms, private enterprises' production activities tend to overlook the cost of environmental pollution and extract environmental dividends to the extreme; given the identical regulatory intensity and environmental regulations, private enterprises, compared to SOEs, will also sacrifice environmental investment for survival and operational pressure. Hence, in the case of greater credit support for private enterprises, environmental pollution is intensified. The primary and secondary coefficients of EXP were significantly positive and negative,

respectively, suggesting an inverted "U-shaped" relationship between environmental pollution and environmental expenditures. At a low level of environmental expenditures, this approach is not conducive to curbing environmental pollution. Beyond a certain limit, environmental expenditures can effectively diminish the level of environmental pollution. Possible reasons for this are analyzed as follows. EXP in this paper indicates the ratio of government environmental expenditures to total budget expenditures. Productive government expenditures can be directed toward the production and reproduction of the economy and can contribute to economic progress. Other nonproductive expenditures, such as government spending on education, science, and technology, are ultimately targeted at economic development; some studies have also offered more direct evidence that government expenditures facilitate economic growth. As a by-product of economic development, environmental pollution is also affected by government expenditures. At a lower proportion of environmental expenditures, other government expenditures support economic development and thus expedite environmental pollution, While the expenditures structure undergoes a change with environmental expenditures exceeding the critical value, government interventions in the environment are further heightened. In parallel, other government expenditures are restricted, and thus economic development is suppressed, reducing environmental pollution and achieving improved environmental quality.

*4.2. Empirical Analysis of Spatial Econometrics with Interaction Terms*

Because of their inferiority to SOEs in terms of asset scale and solvency, private enterprises are subject to credit restrictions by banks owing to credit security, in which the government plays an important role. On the one hand, corporate tax revenue constitutes a key source of government financing, so the government will limit the credit support of state-owned banks for private enterprises due to tax stability considerations. On the other hand, local governments will control the operating behavior of local financial institutions through equity participation and leverage urban commercial banks as a supplement to state-owned banks' funds, further manipulating credit fund allocation across private enterprises and SOEs. Thus, it is of relevance to study the interaction effect of credit allocation structure and government environmental expenditures. Traditionally, the product cross term is introduced directly into the regression, which apparently gives rise to multicollinearity. Pursuant to Aiken and West [47], it was feasible to obtain the interaction term by decentering and then multiplying the two main terms before adding them to the regression equation, which can prevent a distortion of the main effect coefficients due to multicollinearity with the regression results shown in Table 4.

As revealed by the results shown in Table 4, the tests on model selection were comparable to those shown in Table 3. The SEM model controlling for temporal fixed effects offered a stronger explanation for the sample. Subsequent analysis was primarily derived from the results of this model. The coefficient of CAS was positive and passed the 5% significance test. The primary coefficient of EXP was greater than zero and passed the 10% significance test, while the secondary coefficient was less than zero but not significant. These results were comparable to the test results shown in Table 3, suggesting a robust analysis and argumentation against the results listed in Table 3. There was a significant positive coefficient on the interaction term, indicating a positive marginal effect of credit allocation structure on environmental pollution, given a fixed government environmental expenditure. As government environmental expenditure increases, the marginal effect of credit allocation structure on environmental pollution increases. In other words, an increased proportion of credit resources taken up by private enterprises leads to intensified environmental pollution at a certain level of government environmental expenditure; as government environmental expenditure increases, the larger proportion of credit resources taken up by private enterprises leads to heavier environmental pollution, which seems to be inconsistent with the attempt. It may be that the government takes on more responsibility for environmental conservation when the proportion of government environmental expenditure increases. For private enterprises, it may be that since the government has assumed more respon-

sibility for environmental protection, they are likely to reduce their own investment in environmental protection in their own interest, leading to an imbalance between governmental and corporate environmental responsibilities. The phenomenon may also be attributed to the proportion of government environmental expenditures that are corporate environmental subsidies. When the proportion of government environmental expenditure increases, enterprises receive more environmental subsidies. Some enterprises may divert environmental subsidies for their own production expansion, leading to an increase in their pollution emissions and a marginal effect on environmental pollution. Further analysis revealed that before the introduction of the interaction term, the critical value of the EXP inflection point was 0.0377, while that after the introduction of the interaction term was 0.0535. As a result, private enterprises have greater access to credit resources, which is not conducive to the government's efforts to tackle environmental pollution, hindering the effectiveness of government environmental expenditures on environmental pollution.

**Table 4.** Spatial econometric results (including interaction terms).

| Variable | SAR Model | | | | SEM Model | | | |
|---|---|---|---|---|---|---|---|---|
| | nonF | sF | tF | stF | nonF | sF | tF | stF |
| constant | 0.0067 (1.45) | | | | 0.0198 *** (4.91) | | | |
| CAS | 0.0056 *** (3.08) | −0.0007 (−0.56) | 0.0023 (1.28) | −0.0006 (−0.46) | 0.0063 *** (3.42) | −0.0006 (−0.51) | 0.0045 ** (2.55) | −0.0006 (−0.46) |
| EXP | 0.2915 * (1.96) | −0.0684 * (−1.83) | 0.1973 (1.42) | −0.0722 * (−1.91) | 0.2764 ** (2.22) | −0.0531 (−1.52) | 0.2461 * (1.92) | −0.532 (−1.49) |
| $EXP^2$ | −2.1938 (−0.98) | 0.8034 (1.58) | −1.0451 (−0.51) | 0.8439 (1.63) | −2.8765 (−1.54) | 0.6355 (1.32) | −2.3021 (−1.19) | 0.6285 (1.28) |
| $CAS\_C * EXP \_C$ | 0.9954 *** (4.76) | −0.0311 (−0.53) | 0.8979 *** (4.61) | −0.0266 (−0.44) | 0.6482 *** (3.65) | −0.0378 (−0.66) | 0.6759 *** (3.71) | −0.0399 (0.68) |
| INV | −0.0032 ** (−2.26) | 0.0002 (0.35) | −0.0104 *** (−5.97) | −0.0002 (0.32) | −0.0105 *** (−6.73) | 0.0001 (0.01) | −0.0119 *** (−7.21) | 0.0001 (0.01) |
| INN | 0.0518 *** (5.19) | 0.0053 (1.17) | 0.0528 *** (5.57) | 0.0077 (1.37) | 0.0571 *** (5.85) | 0.0071 (1.41) | 0.0547 *** (5.66) | 0.0078 (1.38) |
| INDU | 0.0181 *** (5.44) | 0.0018 (0.98) | 0.0291 *** (8.15) | 0.0009 (0.32) | 0.0198 *** (6.23) | 0.0015 (0.79) | 0.0248 *** (7.47) | 0.0003 (0.12) |
| CITY | −0.0096 ** (−2.04) | 0.0021 (0.57) | −0.0123 *** (−2.81) | 0.0029 (0.47) | −0.0101 ** (−2.37) | 0.0037 (0.93) | −0.0109 ** (−2.55) | 0.0065 (0.99) |
| PGDP | 0.0002 (0.68) | 0.0001 (0.74) | −0.0003 (−1.09) | 0.0001 (0.89) | −0.0003 (−1.36) | 0.0001 (0.59) | −0.0005 * (−1.94) | 0.0002 (0.94) |
| ρ | 0.1929 *** (2.65) | 0.3429 *** (4.31) | 0.2529 *** (3.84) | 0.4042 *** (5.34) | | | | |
| λ | | | | | 0.6249 *** (10.73) | 0.3589 *** (4.52) | 0.4409 *** (5.97) | 0.4118 *** (5.43) |
| $R^2$ | 0.4659 | 0.9855 | 0.5566 | 0.9857 | 0.2983 | 0.9837 | 0.5038 | 0.9839 |
| corr $R^2$ | 0.4291 | 0.0628 | 0.5156 | 0.0783 | 0.364 | 0.0469 | 0.5055 | 0.0585 |
| Log L | 908.254 | 1295.855 | 926.812 | 1297.741 | 922.032 | 1295.564 | 931.689 | 1297.215 |

Note: *, **, and *** indicate that significance tests at the 10, 5, and 1% levels were passed, respectively, and the t-statistic values are shown in parentheses.

### 4.3. Robustness Test

To verify the robustness of the empirical results, CAS2 was selected as a substitution variable for the credit allocation structure to conduct a regression. Adhering to the above empirical approach, the test results are presented in Table 5. Due to length constraints, according to the model selection test results, only the SEM model with stronger temporal fixed effects explanations for the sample and the OLS estimation results used for the model selection test are presented in Table 5.

**Table 5.** Robustness test.

| Variable | Without Interaction Terms | | With Interaction Terms | |
|---|---|---|---|---|
| | OLS | SEM | OLS | SEM |
| constant | 0.0096 ** | | 0.0132 *** | |
| | (2.13) | | (2.87) | |
| CAS2 | 0.0003 | 0.0012 ** | 0.0003 | 0.0011 ** |
| | (0.51) | (2.57) | (0.47) | (2.44) |
| EXP | 0.5148 *** | 0.2751 ** | 0.4265 *** | 0.2337 * |
| | (3.21) | (2.12) | (2.66) | (1.83) |
| EXP$^2$ | −6.64 *** | −3.7565 ** | −4.7832 ** | −2.6405 |
| | (−2.85) | (−2.01) | (−2.01) | (−1.41) |
| CAS2_C $*$ EXP | | | 0.2083 *** | 0.1626 *** |
| | | | (2.92) | (2.94) |
| INV | −0.0037 ** | −0.0129 *** | −0.0042 *** | −0.0137 *** |
| | (−2.39) | (−7.69) | (−2.76) | (−8.24) |
| INN | 0.0596 *** | 0.0606 *** | 0.0651 *** | 0.0665 *** |
| | (5.79) | (6.27) | (6.33) | (6.87) |
| INDU | 0.0209 *** | 0.0279 *** | 0.0182 *** | 0.0253 *** |
| | (5.95) | (8.87) | (5.07) | (7.89) |
| CITY | −0.0089 | −0.0077 | −0.0144 ** | −0.0121 ** |
| | (−1.57) | (−1.56) | (−2.44) | (−2.39) |
| PGDP | 0.0003 | −0.0007 *** | 0.0005 | −0.0006 ** |
| | (0.81) | (−2.68) | (1.41) | (−2.21) |
| λ | | 0.4989 *** | | 0.5079 |
| | | (7.19) | | (7.39) |
| Robust LMsar | 6.5004 ** | | 4.3054 ** | |
| Robust LMerr | 20.5331 *** | | 14.8598 *** | |
| Time fixed effect | | control | | control |
| R$^2$ | | 0.4418 | | 0.4642 |
| corr R$^2$ | | 0.4469 | | 0.4695 |
| Log L | | 924.955 | | 929.317 |
| test | Statistics | *p*-value | Statistics | *p*-value |
| Moran | 0.219 | 0 | 0.194 | 0 |
| Walds | 409.461 | 0 | 432.422 | 0 |
| Lratios | 47.597 | 0 | 46.361 | 0 |
| LMsar | 145.112 | 0 | 110.839 | 0 |
| LMerr | 20.658 | 0 | 16.112 | 0 |

Note: *, **, and *** indicate that significance tests at the 10, 5, and 1% levels were passed, respectively, and the t-statistic values are shown in parentheses.

According to the results shown in Table 5, multiple tests for spatial correlation indicated a significant spatial correlation in the spatial data after substituting the credit allocation structure substitution variables, requiring the adoption of a spatial econometric model. For the model identification tests, the Robust LMerr test was more significant for both models with and without the interaction term, so an SAR model should be used. Spatial correlation coefficient λ was significantly positive in both SEM models, suggesting that replacing the explanatory variables does not alter the spatial dependence of environmental pollution and its influencing factors; the CAS2 indicator was constructed from the ratio of long- and short-term loans of private enterprises to those of SOEs of listed companies. The significant positive coefficients of CAS2 shown in all of the regression results indicated that as private enterprises receive more bank credit support, this is likely to have a tendency to exacerbate environmental pollution. In comparison to most small and medium-sized enterprises (SMEs), listed companies generally enjoy a certain scale and sound operating conditions (ST enterprises were excluded from the data). Nonetheless, private listed companies still assume less social and environmental responsibility than SOEs, a finding that is largely consistent with the previous empirical results [20,48]. Regarding the inverted "U-shaped" relationship between EXP and the explained variables, in the SEM model with

the interaction term, its coefficient was significantly positive, suggesting that government environmental expenditures can mitigate the impact of corporate production on the environment to some extent. Moreover, in the regression results using CAS, the coefficient of the interaction term was 0.6759***, outweighing that of the regression results using CAS2 at 0.1626***. Similar results were found for the coefficients of CAS and CAS2 irrespective of whether the interaction term was included. The larger coefficient of CAS than that of CAS2 indicated that listed industrial enterprises contribute less to environmental pollution than general industrial enterprises. A possible explanation for this phenomenon is related to the large scale of most listed companies, which pursue long-term benefits of production. Generally, such firms are equipped with sound production systems and are capable of properly handling the pollutants generated during the production process to reduce their environmental hazards; on the other hand, government regulation of listed companies is also more stringent, causing companies to follow environmental regulations to a higher degree.

## 5. Conclusions and Policy Implications

### 5.1. Conclusions

From the perspectives of bank credit allocation and government environmental expenditures, this paper explored the impact of the credit allocation structure between private enterprises and SOEs and government environmental expenditures on provincial environmental pollution. A spatial econometric model was utilized to empirically test the panel data of 31 provincial administrative units in China for 2011–2017. The study found that:

(1). Overall, spatial correlation tests such as Moran's I test show a significant positive spatial correlation between provincial pollution levels in China. This conclusion is in line with prior studies [49,50]. The results also suggest that environmental pollution problems have significant spillover effects in China, which means that pollutants in one area may do more harm to environmental quality by spreading to other areas through the exchange of substances. The result echoes what previous literature has indicated.

(2). Private enterprises allocated more credit capital are not conducive to improving provincial environmental quality (Hypothesis 1 is supported), which suggests that SOEs often take more environmental responsibility than private enterprises. The result echoes previous literature stating that SOEs are expected to have a higher level of social environmental responsibility [51–54]. The robustness test also showed that this phenomenon exists in listed companies, but the impact of listed industrial enterprises on environmental pollution is less significant than that of the general industrial enterprises of above the designated size.

(3). There is an inverted "U-shaped" relationship between government environmental expenditures and environmental pollution levels (Hypothesis 2 is supported). This result indicates that when the proportion of government environmental expenditures is low, increasing government environmental expenditures may not control environmental pollution, and when the proportion of government environmental expenditures exceeds a certain threshold, increasing government environmental expenditures can improve environmental quality.

(4). The interaction term coefficient of credit allocation structure and government environmental expenditures is positive, which means that the marginal effect of credit allocation on environmental pollution increases with an increase in the proportion of environmental expenditures (Hypothesis 3 is supported). The threshold of the inflection point of the inverted "U-shaped" relationship between environmental expenditures and environmental pollution increases after the introduction of the interaction term. This result shows that competition for credit resources between SOEs and private enterprises is not conducive to the improvement of environmental pollution, diminishing the effect of government environmental expenditures on environmental pollution control.

### 5.2. Policy Implications

Based on the above findings, several policy implications are provided as follows:

First, environmental protection policies with regional coordination should be established, which will require the development of synergistic provincial collaborative environmental protection regulations to prevent the formation of environmental protection loopholes caused by structural differences between regions in environmental regulations. Such policies will also involve reinforcing the cooperation of environmental protection between provinces, removing barriers to environmental protection, and accelerating the integration of regional protection efforts.

Second, prior literature has indicated that enterprises' social environmental responsibility could enhance the innovation [55]. Our study focused on private enterprises and we found that private enterprises need to increase their environmental responsibility more. The government should strengthen the environmental protection supervision of private enterprises, improve relevant laws and regulations, and let more private enterprises assume the responsibility for environmental protection. At the same time, the government should guide and encourage enterprises to develop a green economy and cleaner production technologies to reduce the environmental cost of the production process and accordingly provide certain tax incentives and financing convenience for enterprises developing green industries or clean technologies.

Third, the regression results of four SEM models controlling the time fixed effect in Tables 4 and 5 showed critical values of environmental expenditures of 0.038, 0.053, 0.037, and 0.044, respectively. As shown in Table 1, the average proportion of environmental expenditures in each province of China is 0.029, which indicates that the level of environmental expenditures in China is still temporarily lower than the critical value and that the effect of environmental expenditures on environmental pollution control and environmental quality improvement is still limited. In recent years, the proportion of environmental expenditures in most provinces and cities in China has been rising each year, but it is still insufficient on the whole. It is necessary for the government to continue to increase the proportion of environmental expenditures, and at the same time, attention should be given to the application quality of environmental expenditures. Targeted environmental expenditures projects should be set up to address some key problems.

Fourth, governments' fiscal and monetary policies should work together. According to environmental pollution in a given region, bank credit allocation and government environmental expenditures should be adjusted in a targeted manner to achieve win–win outcomes for economic development and environmental quality.

**Author Contributions:** J.H. and T.L. conceived and designed the research and edited the paper; Q.Y. analyzed the data and wrote the paper; Q.Y. and Z.Z. reviewed and edited the paper. All authors have read and agreed to the published version of the manuscript.

**Funding:** The authors gratefully acknowledge the support from National Natural Science Foundation of China (Grant No.71573240,71731010).

**Institutional Review Board Statement:** Not applicable.

**Informed Consent Statement:** Not applicable.

**Data Availability Statement:** Data available on request.

**Conflicts of Interest:** The authors declare no conflict of interest.

### Appendix A

Calculation method of the Comprehensive Index of Environmental Pollution (P):
First: Standardize the original pollutant data:

$$K''_{ij} = \frac{Z_{ij} - \min(Z_{ij})}{\max(Z_{ij}) - \min(Z_{ij})}$$

where $Z_{ij}$ represents the value of pollutant indicator $j$ of province $i$, $i \in [1, n]$, and $j \in [1, m]$.

Second: Perform coordinate translation on the standardized data: $K'_{ij} = 1 + K''_{ij}$.

Third: Calculate the proportion of pollutant indicator $j$ in province $i$:

$$K_{ij} = \frac{K'_{ij}}{\sum_i^n K'_{ij}}.$$

Fourth: Calculate entropy $e_j$ and coefficient of variation $g_j = 1 - e_j$ of pollutant indicator $j$:

$$e_j = -\frac{1}{\ln(n)} \sum_{i=1}^{n} K_{ij} \ln(K_{ij}).$$

Fifth: Calculate the weight of pollutant indicator $j$ in the comprehensive pollutant index:

$$W_j = \frac{g_j}{\sum_{j=1}^{m} g_j}.$$

Sixth: Calculate the comprehensive pollutant index:

$$P_i = \sum_{j=1}^{m} W_j K_{ij}$$

where $P_i$ represents the comprehensive pollutant index of province $i$. The larger the value is, the worse environmental pollution in the province is.

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
