# Peer review of "Environmental Effects of Credit Allocation Structure and Environmental Expenditures: Evidence from China"

_sustainability, doi:10.3390/su13115865_

Round 1
Reviewer 1 Report
Governmental intervention in ecological economics is unavoidable, and the related effects need better understanding. The reviewed manuscript provides with an interesting example from China. The authors attempt to examine the environmental effects of credit allocation structure and environmental expenditures. The study itself deserves attention, and many findings are really important to communicate them to the international audience. Nonetheless, the manuscript needs various improvements.
- The first general concern: please, explain whether you have considered all or only selected enterprises. Evidently, not all enterprises pollute environment. So, how this is addressed in your study?
- The second general concern: it would be good to read more about whether the analyzed issues are specific to China or important to some (or many) other countries.
- What does mean comma in the beginning of the authors' list? Any name is missed? Please, check carefully!
- Key words: use words different from those used in the title.
- I think that Introduction is too lengthy. You need to split it into 'Introduction' (there, please, put your problem into the frame of international, not only China-related studies and give your objective) and 'Conceptual Background' (you need to consider a bigger amount of sources).
- Your section 3 should be titled 'Materials and Methods'. Start with your data and then go to your methodology. All used sources of data should be cited properly.
- Table 2: please, give full names of your variables, in addition to acronyms.
- All your findings should be gathered in the section 'Results'.
- You need to write a new section 'Discussion' offering generalization and explanation of your findings, as well as their comparison to the outcomes of some other research (not necessarily related to China). Please, do not forget to include the main conclusions from these interpretations to your section 'Conclusions'.
- Lines 545-546, 561-562: please, give citations to these previous studies.
- I suggest to discuss whether your findings are meaningful to only China or they are of broader importance.
- I think considering and citing this paper may be helpful: https://www.sciencedirect.com/science/article/pii/S0959652620310751
- Generally, I think the list of references is inadequately short for such a paper. Please, think about adding up to 20 fresh and relevant sources.
- I suggest to polish your writing for making it clearer and easier-to-read. I also ask you to distinguish better between your assumptions, previous research findings (citations, please!), and your findings (please, give numbers or refer to tables and figures).
Author Response
请参阅附件

Reviewer 2 Report
Dear author(s)
Please see the attached pdf.file to check a peer review result.

Round 2
Reviewer 1 Report
The authors have done a lot of work, and the revised version of their manuscript looks much better. This in-depth study addresses an urgent research question and offers its appropriate solution. The outcomes are novel and internationally important. The number of literature sources is appropriate.
Nonetheless, all this manuscript is perfect scientifically, I see some technical issues to be improved:
1) The language needs to be brought in order. For instance, Line 11: Resent literature argue -> The recent literature argues.
2) The outline should be provided with more accuracy. For instance, I see some tables without captions.
3) Section 5 is DIscussion and it should be titled as Discussion. If so, 5.1 should be titled anyhow differently. Conclusions are absent in this section. If the authors wish (I do not insist), they can create a new section 6 entitled Conclusions and there give a list of 3-5 main findings.
Author Response
The comments you put forward have been properly revised. Section 5.1 includes the main conclusions of the article, and after each conclusion, a comparison and discussion between the conclusion and the relevant conclusions of the existing literature are followed. Therefore, we don't create a new section to list the conclusions. Section 5.1 is renamed as 'Conclusions'. In addition, we have polished the English grammar and expressions in the manuscript as we can.
Thank you for your careful review and remarkable comments. Your suggestions are impressive and help us a lot.
Sincerely
Reviewer 2 Report
Dear Authors
I have a good impression of this revision.
This is because you decide to delete the Model Section after you recalculate suggested model's setting and recheck obtained results.
Moreover, specializing empirical analysis enhances results and implications of this study, which are accepted by other referee with empirical skill.
Thus, I recommend the editor to accept publishing this paper.
Sincerely.
Author Response
Thank you for your careful review and remarkable comments. Your suggestions of the model revision are impressive and help me a lot.
Sincerely